# T-Type Calcium Channels: A Mixed Blessing

**DOI:** 10.3390/ijms23179894

**Published:** 2022-08-31

**Authors:** Dario Melgari, Anthony Frosio, Serena Calamaio, Gaia A. Marzi, Carlo Pappone, Ilaria Rivolta

**Affiliations:** 1Institute of Molecular and Translational Cardiology, IRCCS Policlinico San Donato, Piazza Malan 2, 20097 San Donato Milanese, Italy; 2School of Medicine and Surgery, University of Milano-Bicocca, Via Cadore, 48, 20900 Monza, Italy; 3Arrhythmology Department, IRCCS Policlinico San Donato, Piazza Malan 2, 20097 San Donato Milanese, Italy; 4Faculty of Medicine and Surgery, University Vita-Salute San Raffaele, Via Olgettina 58, 20097 Milan, Italy

**Keywords:** T-type Ca^2+^ channel, T-type calcium channel blocker, cancer therapy, cardiotoxicity, peripheral neuropathy, mibefradil, bortezomib, carfilzomib

## Abstract

The role of T-type calcium channels is well established in excitable cells, where they preside over action potential generation, automaticity, and firing. They also contribute to intracellular calcium signaling, cell cycle progression, and cell fate; and, in this sense, they emerge as key regulators also in non-excitable cells. In particular, their expression may be considered a prognostic factor in cancer. Almost all cancer cells express T-type calcium channels to the point that it has been considered a pharmacological target; but, as the drugs used to reduce their expression are not completely selective, several complications develop, especially within the heart. T-type calcium channels are also involved in a specific side effect of several anticancer agents, that act on microtubule transport, increase the expression of the channel, and, thus, the excitability of sensory neurons, and make the patient more sensitive to pain. This review puts into context the relevance of T-type calcium channels in cancer and in chemotherapy side effects, considering also the cardiotoxicity induced by new classes of antineoplastic molecules.

## 1. T-Type Calcium Channels

Transient T-type or Low-Voltage Activated (LVA) calcium (Ca^2+^) channels are voltage-dependent ion channels that open at relatively low membrane potentials (i.e., between −70 to −60 mV, with maximum peak current between −30 to −10 mV), allowing extracellular calcium to enter the cell, and which rapidly inactivate (i.e., tau of 15–30 ms) and slowly deactivate [1,2]. The low threshold of activation, not far from the resting potential of most excitable cells, together with their fast kinetics, makes T-type Ca^2+^ channels key modulators of cellular excitability and pacemaking [2,3]. Moreover, T-type Ca^2+^ channels have a voltage range of activation that overlaps that of their steady-state inactivation, meaning that, over a small near-resting voltage range, a fraction of channels can open without completely inactivating, generating a “window” current, distributed around −60/−50 mV, that modulates intracellular calcium levels [4]. T-type Ca^2+^ channels are involved in multiple physiological processes, such as neuronal firing, nociception, electrical automaticity, blood vessel constriction and dilation, lymphatic vessel pacemaking and contraction, smooth muscle contraction, myoblasts fusion, neurotransmitter release, fertilization, cell growth, differentiation, and proliferation [2,5,6,7]. Thus, they are expressed in a variety of excitable and non-excitable tissues in which they display distinctive behaviors at the pharmacological and kinetic (especially in terms of inactivation) levels. This is partly due to the differential and heterogeneous expression of the following three independent T-type channel genes that encode, respectively, for the three alpha subunit subtypes named Cav3.1, Cav3.2, and Cav3.3: *CACNA1G*, *CACNA1H*, and *CACNA1I* [8,9,10]. Similar to other Ca^2+^ channels, like the long-lasting L-type High-Voltage Activated (HVA) ones (i.e., Cav1.x and Cav2.x), the Cav3.x alpha pore-forming subunit is organized into four domains (DI-DIV), each formed by six transmembrane segments (S1–S6), where the S4 segment contains multiple positively charged arginine or lysine residues that serve as a voltage sensor, and with the pore comprised of between segments S5 and S6. This region, responsible for selective permeability, contains four key acidic glutamate or aspartate residues [11]. In particular, the selectivity filter is determined by two glutamate residues in domains I and II and two aspartate residues in domains III and IV, a structure that differs from the one composed of four glutamate residues found in HVA Ca^2+^ channels [12]. In contrast to Cav1.x and Cav2.x, Cav3.x alpha channels do not have either the alpha-interaction domain (AID), an 18-residue sequence in the I-II intracellular linker loop necessary for the interaction with the beta subunit and conserved among all HVA Ca^2+^ channels, nor the IQ calmodulin-binding motif (“IQ” derives from the first two conserved residues of the motif itself) located in the cytoplasmic C-terminal tail and which binds calmodulin. Moreover, LVA Ca^2+^ channels do not seem to co-assemble with ancillary subunits, and expression of just the Cav3.x alpha is enough to recapitulate the native T-type current waveforms [9]. Within the family of T-type Ca^2+^ channels, Cav3.1 and Cav3.2 can be distinguished from each other through their different sensitivities to nickel inhibition and by their kinetics of recovery from inactivation, while Cav3.3 is characterized by much slower kinetics of activation and inactivation [13,14]. Another level of complexity is given by multiple alternative splicing variants that differ at both pharmacological and electrophysiological levels [15]. Despite functional and pharmacological differences, subtype-specific experimental tools (e.g., inhibitors) are still lacking (the ones available are summarized in Table 1), making the study of T-type Ca^2+^ channels in native tissues and cells particularly intricate [3,15], but still necessary for a complete understanding of their physiological and pathophysiological role.

## 2. T-Type Ca^2+^ Channels in the Heart

In the heart, T-type Ca^2+^ channels had been traditionally considered a minor player in cardiac calcium handling. In fact, the vast majority of calcium influx responsible for cardiomyocytes contraction is managed by the more abundantly expressed HVA Ca^2+^ channels. This view has developed over the last 30 years, and, nowadays, cardiac T-type Ca^2+^ channels are considered key regulators of cardiac automaticity, development, and excitation-contraction coupling in several animal models, including mouse, rat, cat, pig, and dog [2,31]. At the cardiac level, T-type Ca^2+^ current (ICa_T_) is carried mainly by the Cav3.1 and Cav3.2 sub-types [31,32]. Their expression in cardiac tissue reaches a maximum in embryonic development and dramatically falls in the post-neonatal phase [31,32]. In particular, the amount of T-type Ca^2+^ channels decrease by about 80% from the embryonic stage to adulthood [33]. During fetal development, Cav3.2 is the most abundant sub-type expressed throughout the heart [34]. In the perinatal stage, the expression of Cav3.2 starts to decrease, while Cav3.1 levels rise and become the predominant adult cardiac sub-type [35]. In the adult heart, T-type Ca^2+^ channels are not expressed in ventricular myocytes, and tend to localize in the conduction system, and in all cell types, characterized by automaticity, where they exert a pacemaker role and function in the depolarization of the sinoatrial nodal cells. The Cav3.1/Cav3.2 ratio varies between different animal models, probably underlying the distinctive heart rates of different mammalian species [31]. Moreover, an inverse correlation has been described between sinoatrial ICa_T_ amplitude and body size, with smaller animals exhibiting a more prominent T-type current [1]. Despite this body of evidence, ICa_T_ has never been directly recorded in human nodal cells [1,36]. On the other hand, transcripts of both Cav3.1 and Cav3.2 have been found in the human sinoatrial node [8,37], with only Cav3.1 detected at the protein level [38]. Some evidence suggests a functional role of ICa_T_ in humans, as oral administration of mibrefradil, a relatively selective T-type Ca^2+^ channel inhibitor, reduced the pacemaker activity of the sinus node [39]. Finally, T-type Ca^2+^ channels are involved in the diseased heart. Indeed, despite not being expressed in healthy adult cardiomyocytes, as already mentioned, an increase in ICa_T_ has been reported in several animal models of heart failure and cardiac hypertrophy [32,40,41]. A greater expression of ICa_T_ can lead to alteration of intracellular calcium handling, intracellular calcium accumulation, and unbalanced calcium signaling. In fact, as demonstrated by knock-out mice, Cav3.2 is involved in the cardiac hypertrophic response, either mediated by mechanical stress, pressure overload, or angiotensin II infusion [42].

## 3. T-Type Ca^2+^ Channels in Pain Modulation and in Chemotherapy-Induced Peripheral Neuropathy

T-type Ca^2+^ channels were first described in peripheral sensory neurons whose cell bodies are located in the dorsal root ganglia (DRGs) [43]. DRGs are key sites for the mechanism underlying chronic and/or neuropathic pain perception [44]. Within this context, T-type Ca^2+^ channels, and in particular the Cav3.2 sub-type, are key players in the acute nociceptive processing induced by reducing agents [45] and are also associated with chronic pain symptoms in rats with peripheral axonal injury [46]. Interestingly, Cav3.2 seems to be particularly highly expressed in a subpopulation of nociceptive, capsaicin-sensitive DRG neurons (called “T-rich”) which exhibit T-type, but not L-type, calcium currents [47]. The role of Cav3.2 in pain perception is confirmed by several studies on different pain models in which channel expression and/or activity are increased after pain-inducing treatments, such as DRG chronic compression, spinal nerve ligation, paclitaxel-induced peripheral neuropathy, and others (for review see [48]). Despite the mounting evidence of an association, the mechanism underlying the increase of Cav3.2 in pain models remains elusive. There is a general inconsistency among studies focused on changes in total protein expression, as some reported an increase while others have observed no change. More agreement is found regarding surface protein expression which is suggested to be augmented in both early and late phases of chronic pain [49,50]. At least, in the latter, this is thought to be related to reduced internalization, due to lower levels of ubiquitination as a direct consequence of the overexpression of the ubiquitin-specific cysteine protease 5/isopeptidase (USP5) [51,52], which interacts with the III-IV linker of the Cav3.2 T-type channel, enhancing its stability. 

T-type Ca^2+^ channels are also involved in a specific form of peripheral neuropathy induced by chemotherapy (CIPN). CIPN is a major dose-limiting side effect of several anticancer agents, such as immunomodulatory, platinum-based drugs, vinca alkaloids, epothilones, taxanes, and proteasome inhibitors [53,54]. Immunomodulatory drugs (e.g., thalidomide) are used in the treatment of multiple myeloma. They induce CIPN by downregulating TNF-α and accelerate neuronal cell death. Platinum-based (e.g., oxaliplatin, cisplatin and carboplatin) antineoplastic drugs are widely used in the treatment of several types of solid tumors. Their involvement in CIPN is due to their effect, among others, on the activity of potassium channels, transient receptor potential (TRP) and voltage-gated sodium channels (Nav 1.6, 1.7 and 1.9). Indeed, an increase in Na^+^ conductance and a reduction in the threshold potential and membrane resistance result in hyperexcitability of peripheral neurons [54]. Vinca alkaloids, used in breast cancer, germ cell tumors, Hodgkin and non-Hodgkin lymphomas, osteosarcoma, and neuroblastoma, inhibit the assembly of microtubules and promote their disassembly, thus disrupting axonal transport and leading to metaphase arrest. They are known to alter the expression of ion channels [54]. Epothilones (e.g., ixabepilone), used in the treatment of breast, ovarian, prostate, and non-small cell lung cancer act as tubulin destabilizers, causing impairment of cancer cell division leading to cell death. Meanwhile, they are responsible for the impairment of axonal transport of synaptic vesicles loaded with essential cellular components, including ion channels. Taxanes (e.g., paclitaxel, docetaxel, and cabazitaxel), are used for the treatment of ovarian, breast, non-small cell lung cancer and prostate cancer [55]. Similar to epothilones, they bind to the β-tubulin subunit, stabilizing the microtubule structure and preventing depolymerization. This condition leads to the arrest of the cell cycle at the G2/M phase. Moreover, microtubule stabilization modifies the expression and function of Na^+^, K^+^, and TRP ion channels. In particular, they decrease the expression of potassium channels and increase that of sodium Nav1.7 channels, which results in the hyperexcitability of peripheral neurons. 

Additionally, taxanes exert a direct effect on Cav3.2 and ICa_T_. In fact, Taxol (paclitaxel), the most commonly used taxane, has a >50% probability of inducing peripheral neuropathy, which can become chronic and irreversible in a subgroup of patients [56]. Moreover, Li and colleagues showed that, in neurons isolated from rat DRGs, Taxol increased both Cav3.2 expression and ICa_T_ density [57]. The treatment also left-shifted both the ICa_T_ voltage-dependent activation and the steady-state inactivation curves, increasing the number of available channels and potentially lowering the neuronal firing threshold [57]. 

Another chemotherapeutic agent that causes CIPN through a direct effect on Cav3.2 is the boronic acid dipeptidase 20S proteasome complex inhibitor bortezomib (BTZ). This class of antineoplastic drug has been developed to tackle cancer, since an over-activation of the proteostatic system machinery (e.g., the ubiquitin proteasome- and the autophagy lysosome-degradation systems) is a well-known characteristic of advanced tumors [58,59,60]. By inhibiting proteasome degradation, BTZ elevates Cav3.2 protein levels and the related current in afferent neurons, leading to BTZ-induced peripheral neuropathy (BIPN). BTZ is commonly used in the treatment of multiple myeloma and mantle cell non-Hodgkin’s lymphoma [61] and exerts its therapeutic action by inducing an arrest of the cell cycle, upregulating pro-apoptotic genes, and downregulating key factors of angiogenesis, stroma adhesion, cell proliferation and survival [62,63]. 

Despite BTZ efficacy, BIPN is one of the most severe non-hematological side effects of chemotherapeutic agents against multiple myeloma [61]. The ability of BTZ to inhibit proteasome activity in DRG neurons has been demonstrated in rat and mice models of BIPN [64,65]. In a recent study, Tomita and colleagues showed that in a mouse model of BIPN, the protein expression of Cav3.2 and USP5 was upregulated without increasing mRNA levels, suggesting that BTZ increases Cav3.2 protein level by reducing its proteasomal degradation. In fact, BIPN was reversed by knockdown of Cav3.2 and by the administration of T-type channel blockers, including the state-dependent blocker TTA-2, the state-independent blocker PNG, the PNG-analogue KTt-45, and ascorbic acid, which selectively blocks Cav3.2 but not Cav3.1 and Cav3.3 [66]. Interestingly, another new generation proteasome inhibitor, carfilzomib (CFZ), showed minimal neurotoxicity and fewer and milder off-target effects compared to BTZ [67]. This reduced toxicity is thought to be due to higher selectivity of CFZ for the chymotrypsin-like activity of the ꞵ5 sub-unit of the 20S core particle of the proteasome [68]. On the other hand, CFZ treatment was associated with a 5% incidence of unpredictable cardiovascular events, including congestive heart failure, pulmonary edema, decreased ejection fraction, cardiac arrest, and myocardial ischemia [69]. BTZ therapy itself, though, is not without cardiac side effects responsible for therapy discontinuation [70,71].

## 4. The Ubiquitin-Proteasome System and Proteasome Regulation of Cardiac Ion Channels

The Ubiquitin-Proteasome System (UPS) is one of the major protein degradation systems in eukaryotic cells and it accounts for up to 90% of the degradation of long- and short-lived and abnormal intracellular proteins [72]. Despite the cytosolic localization of its components, the UPS can target proteins from the plasma membrane, nucleus, and even from the ER lumen [73]. The pathway through which a protein undergoes UPS degradation is composed of two distinct events: first, a chain of multiple ubiquitin molecules is covalently attached to the target protein, and second, the tagged protein is transported to the proteasome for degradation. The structure and function of the proteasome have been extensively studied and reviewed [74,75,76,77,78,79] and go beyond the scope of this review. The proteasome and the ubiquitin-activating enzymes are constitutively active. Nevertheless, UPS is finely regulated, as the ubiquitination state of a protein is a dynamic counterbalance between ubiquitination and de-ubiquitination [73]. This machinery is of course involved in the regulation of the surface expression not only of T-type but also of several ion channels. The incubation with the proteasome inhibitor MG132, a structural and functional analog of BTZ that enhances Cav3.2 activity in rat DRGs [51], extended the half-life of cardiac Kv1.5 expressed in COS cells, inducing a significant increase in the protein expression level and current amplitude [80]. The expression of the hERG potassium channel, the product of the human ether-a-go-go related-gene, at the membrane of HEK cells, is also regulated by the UPS system [81,82], and proteasomal inhibition by BTZ, MG132, and other drugs rescued trafficking-deficient LQT2-related and schizophrenia-related hERG channel variants [83,84,85,86]. In addition to calcium and potassium channels, the cardiac Nav1.5 sodium channel is also targeted by the UPS [87]. MG132 increased its protein expression and current density in isolated neonatal rat cardiomyocytes and rescued the Nav1.5 reduction in cardiomyocytes of dystrophin-deficient mdx5cv mice [88,89]. Interestingly, in Schistosoma mansoni parasites, MG132 caused a decreased expression of transcripts of different ion channels, including the HVA Ca^2+^ channels, Ca^2+^-activated potassium channels, and ATP-sensitive potassium channels, an effect opposite to that observed in different animal models [90]. That said, it is not surprising that inhibition of the proteasome machinery leads to alterations in excitability, with deleterious effects on neuronal and cardiac activity.

## 5. Paclitaxel, Bortezomib, and Carfilzomib Cardiotoxicity: A New Field That Needs to Be Explored

It is well established that chemotherapeutics induce cardiotoxicity to the point that the field of cardio-oncology has developed. Although the definition of cardiotoxicity commonly indicates a decline in patients’ cardiac function, the spectrum of cardiac side effects of chemotherapeutic treatment is heterogeneous and includes impairment in ventricular depolarization or repolarization and QT interval alterations, arrhythmia, bradycardia, tachycardia, decreases in left ventricular ejection fraction and fractional shortening, and irreversible congestive heart failure [91]. All of which worsen patient quality of life and increase mortality. Anthracyclines are considered the most common culprit drugs causing chemotherapy-induced cardiotoxicity, (acute events in 0.4–41% of patients and chronic events in 0.4–23%) followed by fluoropyrimidines (3–19%) [92,93]. Taxanes are in third position with an epidemiology of 3–20% cardiotoxic events, the most common of which are arrhythmia and cardiac ischemia. In particular, paclitaxel treatment causes acute or sub-acute bradycardia in 30% of patients, cardiac ischemia in 5% of treated patients [91], heart block, and atrial or ventricular arrhythmias in a smaller fraction of patients (0.5%) and restricted left ventricular pump function, and can provoke chronic cardiotoxicity with clinical symptoms of cardiac insufficiency even decades after the end of treatment [94]. 

BTZ and CFZ cardiotoxicity is still a matter of debate. Clinical data are conflicting, as cardiac events are not clearly related to significant cardiovascular risk factors, such as existing cardiac diseases or co-administration of known cardiotoxic drugs [62,95]. Even if rare, BTZ-associated cardiac events have been reported, and include heart failure (the most common), complete atrioventricular block, atrial fibrillation and other forms of arrhythmias, pericardial effusion, orthostatic hypotension, and ischemic heart disease [62,70,71]. CFZ is considered more cardiotoxic than BTZ and it has also been associated with a higher incidence of cardiac arrhythmias [96]. These uncommon events may suggest a mild effect of BTZ and CFZ on the cardiac tissue, that can become life-threatening in the presence of cardiac risk factors and/or compromised substrates.

Animal models have been used to investigate the mechanism behind BTZ and CFZ alleged cardiotoxicity. In male Wistar rats, the administration of BTZ led to left ventricular contractile dysfunction with impaired cardiomyocyte contractility, due to mitochondrial alteration, and reduced ATP production [97]. Moreover, Hasinoff and colleagues recently tested both BTZ and CFZ on primary neonatal rat cardiomyocytes showing that the two compounds induced cell damage at sub-micromolar concentrations. The study argued that the proteasomal inhibition within a cellular environment, characterized by elevated sarcomeric protein turnover, led to cellular damage and subsequent cell death and apoptosis [98]. In another study, Tang and colleagues pointed to an overactivation of the hypertrophy-related calcineurin and nuclear factor of activated T-cells (NFAT) signaling pathway as the culprit for BTZ cardiotoxicity in cultured and in vivo murine cardiomyocytes. In particular, the administration of BTZ induced left ventricular hypertrophy, heart failure, and premature death [99]. 

Despite the evidence, the mechanisms behind BTZ and CFZ cardiotoxicity remain elusive. As reported above, BTZ has a direct effect on the Cav3.2 level of expression in rat DRGs through inhibition of channel ubiquitination and internalization [66]. In the heart, the UPS is the principal protein degradation system, managing the turnover of up to 90% of the cellular proteins [100], and alteration in the UPS can lead to several cardiac diseases, including cardiac hypertrophy, chronic heart failure, and remodeling [101]. It is, therefore, intriguing that to date no studies have been published regarding the potential cardiomyocyte electrophysiological consequences of inhibition of the cardiac proteasome by BTZ and CFZ and the potential of T-type calcium channels as a pharmacological target. To date, several therapeutic strategies and targets have been proposed to reduce clinical cardiotoxicities, among them iron-chelating drugs, β-blockers, renin-angiotensin-aldosterone system (RAAS) inhibitors, SGLT2 inhibitors, late inward sodium current (INa_L_) selective inhibitors, phosphodiesterase-5 inhibitors, metabolic agents, and statins, as well as growth factors and hormones. 

Dihydropyridine Ca^2+^-channel blockers (amlodipine, felodipine) have been suggested as first-line agents in the case of fluoropyrimidine treatments, when chemotherapy-induced cardiotoxicities range from QT prolongation to hypertension and left ventricular dysfunction [102]. Cardiotoxic events were suggested to be mediated by vascular smooth muscle cells and, thus, dihydropyridine Ca^2+^-channel blockers may exert direct vasodilatory effects via the arteriolar smooth muscle. As a side effect, there may be lower extremity edema, the frequency of which is dose-dependent and could be minimized by lowering the dose and by nocturnal administration [103]. On the other hand, non-dihydropyridine Ca^2+^ channel blockers are not indicated, due to drug-drug interactions and for the impact that they may have on the CYP3A4 system, which can lead to increased concentrations of the chemotherapeutic drug. Arterial hypertension is frequently reported (11–45%) in patients receiving VEGF inhibitors, such as bevacizumab and sunitinib. Ca^2+^ channel blockers are usually prescribed in these cases. In contrast, they should be used with caution in cases of arrhythmias, either supraventricular or ventricular, and in particular in bradyarrhythmias [96].

A recent review that summarized therapy-specific cardioprotective strategies only mentioned the chemotherapeutic class of proteasome inhibitors, confirming that, even though some information is available on how to treat these cardiotoxicities, robust data on primary cardioprotective strategies are lacking [104]. Indeed, compared to older classes of anti-cancer agents, proteasome inhibitors have only recently been introduced in clinical practice (the progenitor was approved by the FDA approval in 2008). This could be the main reason why an adequate estimation of their cardiotoxic effects is missing, and why the cellular and molecular mechanisms mediating the cardiotoxicity, and the role of T-type calcium channels, is still poorly explored. It appears clear though, that with the advancement of precision medicine and with the emerging of new classes of chemotherapy and targeted therapy drugs, there is an urgent need to develop novel strategies to mitigate adverse effects and to reduce clinical and subclinical cardiotoxicity.

## 6. T-Type Ca^2+^ Channels in Cancer

Due to their role in the regulation of cell-cycle progression, the aberrant expression and activity of T-type Ca^2+^ channels (Cav3.2) has been demonstrated and implicated in cancer. Thus, the expression of Cav3.X in cancer has been extensively reviewed [105]. We limit this report to a concise summary of the more evident literature.

According to Human Protein Atlas, 27% of glioblastoma biopsies express Cav3.2, while 82% express Cav3.1. Compared to commonly used established cell lines and the normal brain, Cav3.2 expression is elevated in human glioblastoma and in glioblastoma stem cells that are resistant to radio and chemotherapy, and its deregulation correlates with worsened patient survival. An increase in intracellular calcium, modulated by Cav3.2 expression, has been shown to regulate glioblastoma cell proliferation [106,107]. Cav3.2 is upregulated in stages III and IV of medulloblastoma, the most common pediatric malignant brain tumor, and the level of expression correlates with aggressiveness, the occurrence of metastasis, and worsens clinical outcomes. Patients with high Cav3.2 levels show significantly reduced overall survival rates. Moreover, similar to the data learned from patients, the expression of upregulated Cav3.2 was increased in a mouse transgenic model of medulloblastoma tumor tissues compared to the control mice cerebellum tissues [108]. 

In prostate cancer, in samples harboring a mutant androgen receptor (AR) gene, which are, thus, resistant to the common therapy of AR blockers, the expression levels of Cav3.3 were significantly higher compared to samples negative for AR mutation. A further increase in Cav3.1 and Cav3.2 copy number variation rate was identified in neuroendocrine prostate cancer cells, which were highly metastatic and resistant to all available therapies, suggesting that T-type Ca^2+^ channels play a role in the progression from hormone-naïve to neuroendocrine prostate cancer. Furthermore, the expression of Cav3.1 is associated with a poorer prognosis in earlier stages of the disease [109,110]. Data obtained from MCF-7 and MDA-MB-231 breast cancer cells showed high levels of mRNA for T-type Ca^2+^ channels, both Cav3.1 and 3.2, and expression was associated with hyperproliferation [111]. T-type Ca^2+^ channels are highly expressed in cutaneous melanoma where they play a crucial role in cell viability and induction of cell cycle progression. In particular, a progressive increase in the expression of Cav3.2 was found from normal skin to common nevi, to metastatic melanoma in human samples. Melanoma cells harboring mutations in the B-Raf proto-oncogene serine/threonine kinase (BRAF) gene, which is considered a genetic hallmark of >50% of melanoma, showed higher levels of Cav 3.1 and Cav 3.3 mRNA [112]. Immunocytochemistry, as well as qPCR and western blot analysis, revealed that Cav3.1 expression is also upregulated in the epithelial layers of human samples of Oral Squamous Cell Carcinoma (OSCC), and the expression levels correlated with the pathological grades (I vs II and III) and size of the tumor, and with the proliferative and anti-apoptotic potential. In contrast, the expression of Cav3.1 was markedly negative or weak in oral mucosa and epithelial dysplasia [113].

It is interesting to notice that, similar to T-type Ca^2+^ channels, sodium channels can also regulate cancer cell invasion, and their expression seems to facilitate cell motility, migration and invasiveness of cancer cells, and to correlate with metastatic potential [114,115,116]. This similarity may support speculation about a mechanistic relationship between the two types of ion channels. The ability of cancer cells to invade tissues is related to the presence of invasive structures, called invadopodia, that are functionally and morphologically similar to podosomes, that protrude at the edge of the cells, contact the extracellular matrix, and tune its degradation through metalloprotease activity, thus invading the surrounding tissue. Na^+^ and Ca^2+^ channels blockers have been found to reduce the formation of invadopodia, suggesting a possible common involvement [116,117]. These channels are both implicated in physiological membrane depolarization, and, from the biophysical point of view, they both rely on a window current that guarantees a constant inward flux of cations, even in relatively depolarized conditions (for the voltage-gated sodium channel the window current peaks at about −60 mV and may activate T-type calcium channels). In this sense, they both cooperate in increasing the intracellular calcium concentration that support the formation of invadopodia [116], and, thus, the cancer malignancy.

## 7. Therapeutic Strategies Aiming to Control T-Type Ca^2+^ Expression and Activity

From what has been described so far, it is intuitive that the control of the expression or activity of the T-type Ca^2+^ channels would be extremely advantageous in cancer therapy, whether calcium channels are expressed in cancer cells, or calcium channel expression is increased as a consequence of the antineoplastic therapy, as in CIPN. In the translational research field, several methods, such as drug application, gene silencing, short interfering (or small interfering) RNA, and hairpin RNA, have been developed to reduce the CACNA1.X gene expression. The blockade of the channels through specific drugs or gene silencing lowered the proliferation rate, and dramatically reduced cell viability as cell death and apoptosis were promoted, both in breast cancer and in glioblastoma [106,111] (Table 2).

In terms of cancer therapy, Ca^2+^ channel blockers are administered alone or as adjuvant drugs, synergistically combined with conventional chemotherapy (interlaced, or timed sequential therapy), where they may enhance the effectiveness of the therapy and represent a promising strategy for successful cancer treatment. In general, the idea of the timed sequential therapy is based on the concept that, by using a T-type Ca^2+^ channel blocker, the population of metabolically vulnerable cancer cells in the S-phase can be increased, which sensitizes cells to cytotoxic radiotherapy and chemotherapy (as in many cancers). This strategy was validated in a Phase 1b clinical trial (ClinicalTrials.gov identifier: NCT01480050) in which the T-type calcium channel blocker mibefradil was administered to synchronize the cells in the G1/S phase and, then, following its withdrawal, a standard chemotherapeutic cytotoxic agent temozolomide, active at S phase, was administered to kill the cells. Results were published in 2017, revealing that sequential treatment was safe and met the criteria for further evaluation of this regimen, despite the limitations of the study [131]. Mibefradil is a tetralol-derivative, non-dihydropyridine FDA-approved Ca^2+^ channel blocker, previously marketed for the treatment of hypertension and chronic angina pectoris, but withdrawn from the market worldwide in 1998, due to several reports of pharmacokinetic interactions with other drugs metabolized by CYP3A4 and 2D6. It was the first drug to be marketed as a specific T-type Ca^2+^ channel antagonist [134] as it blocks the T-types 10 to 30 times more potently than L-type Ca^2+^ channels, even though an inhibitory effect on the current flowing through the latter (ICa_L_) in vivo cannot be excluded [31]. It has been repurposed due to the fact it significantly inhibited cell growth and proliferation of glioblastoma stem cells and MCF-7 breast cancer cells. Moreover, it decreased cell viability in PC-3 cells, in an in-vitro model of neuroendocrine prostate cancer and in an in vivo model of glioblastoma, prolonging animal survival [106,109]. However, later reports showed that mibefradil can inhibit other ion channels, including the voltage-gated sodium, Ca^2+^- and volume-activated chloride channels, and potassium (inward rectifier, delayed rectifier, and hERG) channels in the sinus and atrioventricular nodes, slowing conduction velocity [48,135].

Resistance in cancer therapy has been associated with hypoxia and HIF1alpha levels in cancer cells. This is true also for glioblastoma cells, in which hypoxia induced Cav3.2 and HIF1 and 2 expressions. Application of mibefradil not only significantly suppressed HIF1 and 2 expressions, but also inhibited the AKT/mTOR pro-survival pathway associated with cancer, and induced signaling changes related to the induction of cell-cycle arrest and apoptosis [106]. Other specific T-type Ca^2+^ channel blockers tested are the tetralol derivatives NNC 55-0396 and SB-209712. In particular, NNC 55-0396, an analog of mibefradil with higher blood-brain-barrier permeability, is more selective, and it blocks especially Cav3.1 and Cav3.2 with lower non-specific effects on L-type Ca^2+^ channels. Similar to mibefradil, NNC 55-0396 altered mitochondrial function, energy metabolism, induced cellular apoptosis in medulloblastoma cells [135], and caused cell-cycle arrest in a wide range of melanoma cells [136]. In some cases, the concentrations at which tetralols proved to be toxic for cultured cancer cells are higher than those required to block T-type Ca^2+^ channels. Thus, it has been proposed that the cancer cell death induced by tetralols was likely due to off-target actions. For example, in glioblastoma and melanoma, mibefradil and NNC 55-0396 seemed to activate IRE1 alpha (Inositol-Requiring Enzyme 1), and the unfolded protein response (UPR) system, leading to the mobilization of calcium from the endoplasmic reticulum, inducing apoptosis [107,112]. 

Anti-psychotic drugs, such as penfluridol and the structurally similar diphenylbutylpiperidine-derivative pimozide, block the calcium current with about tenfold higher selectivity for LVA over HVA channels, but they were also found to be potent inhibitors of hERG (the rapid component of the delayed rectifier), KvLQTI/minK (the slow component of the delayed rectifier), and Kv1.5 (ultra-rapid delayed rectifier) channels, even with much lower affinity, overall inducing QT interval prolongation [22,137]. Flunarizine, originally prescribed for migraine prophylaxis, is another proven T-type Ca^2+^ channel inhibitor whose selectivity is tissue-related, In smooth muscle cells and cardiomyocytes it does not select between L- and T-type, while in neuronal cells it preferentially blocks the T-type Ca^2+^ channels with an equilibrium dissociation constant K_D_ 4-fold higher [138]. A small cyclic peptide named PnCS1 was found to inhibit the three members of the Cav3 family with the same potency. By using a cryo-EM approach, it was established that the cyclic peptide locates in the central pore, between the selectivity filter and the intracellular gate of the Cav3.1 channel, and molecular dynamics simulations revealed remarkable stability of the interaction [139]. The pyridyl amide TTA-A2 has a 300-fold higher affinity for T-type compared to others channels, revealing it to be a potent blocker, and was demonstrated to be efficient in inducing cell death in lung adenocarcinoma cells, and reducing colony formation, a process at the basis of metastatic colonization. In particular, TTA-A2 treatment not only blocked Cav3.X, but also down-regulated its mRNA expression [140]. Similarly, a novel series of N3- substituted dihydropyrimidines has been developed, two of which, named C12 and C13, are highly selective for Cav3.2 and revealed anticancer activity when applied alone or in combination with cisplatin and etoposide on A549 cell line (lung adenocarcinoma) and on a human breast cancer cell line (MDA-MB 231), showing synergistic activity. In silico studies on the same compounds suggested a low level of cardiac toxicity, as these drugs did not block hERG channels, essential for cardiomyocyte repolarization [126]. 

On the other hand, T-type Ca^2+^ channels may also be a pharmacotherapeutic target to reduce neuronal sensitivity, with the potential for the reversal of chemotherapy-induced peripheral neurotoxicity associated with antineoplastics (see CIPN) [141,142]. Suvecaltamide is another potent and selective modulator of T-type Ca^2+^ channels with a high affinity for the inactivated channel conformation. Its administration in a rat model of CIPN provoked an increase in nerve conduction velocity in caudal and sciatic nerves without interacting with BTZ-induced proteasome inhibition activity, and suggested that suvecaltamide does not block or attenuate BTZ anti-tumor activity [142]. Other recent developments in the discovery of novel classes of T-type Ca^2+^ channel blockers of pain, including their analgesic effects in animal models, and in clinical trials, have been proposed by Snutch and Zamponi in 2017 [16]. Despite this plethora of new molecules, most of the T-type Ca^2+^ channel blockers still interfere with other protein functions, in particular ion channels.

## 8. Conclusions

The role of T-type Ca^2+^ channels have naturally been associated with membrane excitability of the central and peripheral neurons, and of the pacemaker cells in the cardiac and smooth muscle tissues. As their activation is voltage-dependent, they promote action potential firing by amplifying weak depolarizing stimuli, driving the membrane potential towards the excitability threshold of other voltage-gated channels. Nevertheless, thanks to the “power” of the window current, they also contribute also to activating biochemical signals that initiate multiple physiological events, even in non-excitable cells, most notably cancer cells [143]. In this context, they are linked to cell-cycle progression, proliferation, survival, migration, and supporting malignant growth. Thus, their activity has to be finely regulated and several pharmacological T-type Ca^2+^ channel blockers have been developed to be used either in the cardiovascular and the oncology fields as chemotherapeutics or in adjuvant therapy. However, precisely because of the relevant physiological and pathophysiological roles of these channels, and also because the drugs have been revealed not to be thoroughly specific, caution is needed in their use as the final effect could be a mixture of on-target and off-target actions. Besides the studied cardiotoxicity, the risk of pathological arrhythmias is always a consideration. Thus, the full exploration of the therapeutic potential of targeting Cav3 channels would benefit from the development of ligands with high potency and selectivity, and from the implementation of cardiomyocyte electrophysiological studies.

## Figures and Tables

**Table 1 ijms-23-09894-t001:** T-type Ca^2+^ channel blockers.

Drug	Chemical Class/Origin	Preferential Block	Treatment Indication	References
(3R,5S)-31c	Benzodiazepine	Cav3.3 > Cav3.1 > Cav3.3	Absence epilepsy	[16]
A1048400	Diphenylpiperazine	T-type and N-type	Tactile allodynia in capsaicin-induced secondary hypersensitivity (animal model)	[17,18]
A-686085	Diphenylpiperazine	L-, N- and T-type	Tactile allodynia in capsaicin-induced secondary hypersensitivity (animal model)	[18]
ABT-639	Sulfonamide	Cav3.2	Diabetic Neuropathy, failed clinical trials for pain/schizophrenia treatment	[19]
ACT-709478	Heteroaromatic amide	Cav3.1 > Cav3.3 > Cav3.2	Generalized epilepsy (Phase II)	[20]
Amiodarone		Na, K and Ca	Class III Antiarrhythmic agent	[21]
Amlodipine (Norvasc)	DHP	Cav3.2 > Cav3.1 and Cav3.3	High blood pressure and coronary artery disease	[17]
Anandamide	Endocannabinoids	T-type		[16]
Arachidonyl-glycine	Anandamide derivative	T-type		[16]
Aranidipine (Sapresta)	DHP	L- and T-type	High blood pressure	[17]
Azelnidipine (CalBlock)	DHP	L- and T-type	High blood pressure	[17]
Barnidipine	DHP	L- and T-type	Hypertension	[17]
Bay K8644	DHP	L- and T-type		[21]
Benidipine (Coniel)	DHP	L- and T-type	Hypertension	[17]
Bepridil	Diamine	Non selective	Angina	[21]
Compound 10d	Hexane derivatives	T-type, hERG, N-type	Neuropathic pain (animal model)	[16]
Compound 10e	Piperazine derivative	Cav3.1, Cav3.2, Cav3.3. No strong effect on Cav1.2 and Cav2.2	CFA-induced inflammatory pain.	[16]
Compound 9b	Sulfonamides	T-type, N-type > hERG	Cold allodynia, mechanincal pain hypersensitivity (animal model)	[16]
Compound 9c	DHP derivative	Cav3.2 > Cav1.2	Inflammatory pain (animal model)	[16]
Compound series	Hybrids of NMP-7 and TTA-A1	Potent Cav3.2 inhibition	Cav3.2-related neuropathic and inflammatory pain (animal model)	[16]
D888 (devapamil)	Phenylalkylamine derivative	L- and T-type (Cav3.2)	Stress induced ulcer in rats	[21]
Diltiazem	DHP	Non selective	High blood pressure, angina, arrhythmias	[21]
Efonidipine (Landel)	DHP	L- and T-type	Hypertension	[17]
Ethosuximide (Zarontin)	Succimide	Cav3.1	absence epilepsy	[20]
Felodipine	DHP	L- and T-type		[21]
Flunarizine	Diphenyldiperazine derivative	T-type	Neuroepileptic agent	[22]
Fluoxetine (Prozac)	Selective serotonin reuptake inhibitors (SSRI)	Cav3.1, Cav3.2 Cav3.3	Depression	[23]
Haloperidol	Butyrophenone	T-type	Neuroepileptic agent	[22]
Isradipine	DHP	Cav3.2		[21]
Kurtoxin	Scorpion venom	L- and T-type		[24]
KYS05041	3,4-Dihydroquinazoline derivative	T-type	In vitro inhibition of cancer cells growth	[25,26]
KYS05047	3,4-Dihydroquinazoline derivative	Cav3.1, Cav3.2	Effective on neuronal circuits	[17]
KYS-05090S	3,4-dihydroquinazoline derivative	Cav3.1, Cav3.2	Inflammatory and neuropathic pain	[16]
Mibefradil (Posicor)	Phenylalkylamine	L- and T-type, Na and K	Hypertension and angina (withdrawn)	[17,27]
MK-8998 (Suvecaltamide)	Pyridyl amide	T-type	Failed clinical trials for pain/schizophrenia treatment	[19]
ML218		Cav3.1, Cav3.3		[17]
N10 and N12	DHP derivative	T-Type	Inflammatory pain (animal model)	[16]
NCC 55-0396		T-type	Tumor-induced angiogenesis in vitro and in vito	[17]
Nicardipine (Cardene)	DHP	L- and T-type	Hypertension and angina	[17]
Niguldipine	DHP	L- and T-type		[21]
Nimodipine (Nimotop)	DHP	L- and T-type	Cerebral vasospasm, ischemia	[17]
Nisoldipine	DHP	L- and T-type		[21]
NMP-181	NMP-7 derivative	CB2 agonist, T-type blocker	Formalin-induced inflammatory pain (animal model)	[16]
NMP-7	Carbazole derivative	Cannabinoid receptors CB1 and CB2 agonist, T-Type blocker	Formalin-induced inflammatory pain (animal model)	[16]
Penfluridol	Diphenylbutylpiperidines	D2 dopamine receptor antagonist, T-type and L-type blocker	Neuroepileptic agent	[22,25]
Perhexiline		L- and T-type (Cav3.2)	Coronary vasodilator, angina	[21]
Pimozide	Diphenylbutylpiperidines	D2 dopamine receptor antagonist, T-type and L-type blocker	Neuroepileptic agent	[22,25]
ProTx I	Tarantula venom	Cav3.1 (Blocks also NaV)		[28]
ProTx II	Tarantula venom	Cav3.2 (Blocks also NaV)		[28]
RQ-00311610		T-type	Increased bladder capacity in bladder outlet obstruction model	[17]
TH-1177	Chemical synthetic peptide		Human cancer prostate cell proliferation	[25]
Trazodone	Serotonin antagonist and reuptake inhibitors (SARIs)	Cav3.1, Cav3.3	Depression	[29]
TTA-A2		Cav3.1, Cav3.3	Effective on neuronal circuits	[17]
TTA-P2	4-aminomethyl-4-fluoropiperdine derivative	T-type	Antinocipetive agent	[17]
Verapamil	Phenylakylamine	L- and T-type	High blood pressure and angina, supraventricular tachycardia	[30]
VH04		Cav3.1		[17]
Z941/944		T-type		[17]
Z944	Piperazine	T-type	Pain (Phase II)pain	[20]
Zonizamide (Excegran)	Sulfonamide	Cav3.2 (non selective)	Epilepsy	[17,20]

**Table 2 ijms-23-09894-t002:** Known compounds targeting T-type calcium channels studied in pre-clinical research or in clinical trials.

Drug(s)	Cancer Type(s)	Model(s)	Reported Adverse Effects	References
5b, 6b, 6c, BK10040, 8, KYS05090,	Lung adenocarcinoma	A549 cell line	N.A.	[118]
Amlodipine	Human epidermoid carcinoma	A431 cell line	N.A.	[119]
Amlodipine	Uveal malignant melanoma	Cutaneous malignant melanoma cell lines and 3D cultures	N.A.	[120]
Amlodipine and doxorubicin, concomitant treatment	Gastric cancer	AGS and MKN45 cell lines	N.A.	[121]
Amlodipine and gemcitabine, concomitant treatment	Pancreatic ductal adenocarcinoma (PDAC)	Orthotopic Xenografts in mice and mouse model of PDAC	Not reported	[122]
Amlodipine and regorafenib, concomitant treatment	Metastatic colorectal cancer	Human patients	Not reported	[123]
Amlodipine and vincristine, concomitant treatment	Neuroblastoma	SH-SY5Ycell line	N.A.	[124]
Ascorbic Acid	Neuroblastoma-glioma	NG108-15 cell line	N.A.	[125]
C12 and C13 with cisplatin, concomitant treatment	Lung adenocarcinoma and human breast cancer	A549 and MDA-MB 231 cell lines	N.A.	[126]
KYS05041 KYS05042, KYS05043, KYS05046, KYS05047, KYS05048, KYS05055, KYS05056, KYS05057, KYS05065, KYS05080, KYS05085, KYS05089, KYS05090	Lung carcinoma, colon cancer, epidermoid carcinoma, maglignant melanoma, ovarian cancer	A549, HCT-15, KB, SK-MEL-2, SKOV3 cell lines	N.A.	[25,118,127]
KYS05090	Lung adenocarcinoma	Xenograft nude mice	Panting; inanimation; loss of locomotor activity; erosion, Diarrhea; soiled perineal region	[125]
KYS05090, 6a, 6c, 6d, 6f, 6g, 6h	Epithelial ovarian cancer	SK-OV-3 cell line	N.A.	[118]
Methanandamide	Neuroblastoma-glioma	NG108-15 cell line	N.A.	[125]
Mibefradil	Pancreatic cancer	Pancreatic cancer xenografts	Not reported	[128]
Mibefradil	Retinoblastoma, breast cancer,	Y79, WERI-Rb1 retinoblastoma, MCF7 cell lines	N.A.	[25]
Mibefradil	Glioma and neuroblastoma	U87MG, A172, U373, T98G, SNB19, U1242, U251 and SF767, C6, GIC, GliNS1, G179NS, G166NS, U3NNN-M NG108-15 and N1E-115 cell lines; primary glioblastoma cells (GBM-6, GBM-10)	N.A.	[25,125]
Mibefradil	Esophageal carcinoma	KYSE150, KYSE180, TE1, TE8	N.A.	[125]
Mibefradil	Colon cancer	HCT116	N.A.	[125]
Mibefradil	Leukemia	MOLT-4, Jurkat, Ball, HL-60, NB4, HEL, K-562, and U937 cell lines	N.A.	[125]
Mibefradil	Glioma and glioblastome	Xenograft injection in mice	Not reported	[125]
Mibefradil	Ovarian cancer	HO8910, A2780 cell lines	N.A.	[125]
Mibefradil	Ovarian cancer	Xenograft nude mice	Not reported	[125]
Mibefradil + Radiosurgery	Glioblastoma	C6 xenograft in rat	Non reported	[125]
Mibefradil + temozolomide	Glioblastoma	GBM xenograft	Not reported	[125]
Mibefradil and carboplatin, timed sequential therapy	Ovarian cancer	In vivo xenografts (mouse model)	Not reported	[129]
Mibefradil and paclitaxel, timed sequential therapy	Breast cancer	In vivo xenografts (mouse model)	Not reported	[130]
Mibefradil and temozolomide, timed sequential therapy	Glioma	Human patients	Well tolerated (NCT01480050)	[131]
Nickel	Neuroblastoma-glioma	NG108-15 cell line	N.A.	[125]
Nickel	Prostate cancer	LNCaP cell line	N.A.	[125]
Niguldipine	Glioma	GIC, GliNS1, G179NS, and G166NS, U3NNN-MG cell lines	N.A.	[125]
Niguldipine	Glioma	Xenograft injection	Not reported	[125]
NNC 55-0396	Breast cancer	MCF-7 cell line	N.A.	[111]
NNC 55-0396	Human glioblastoma	Tumor xenograft mouse model	No side-effect on liver function	[132]
NNC 55-0396	Leukemia	MOLT-4, Jurkat, Ball, HL-60, NB4, HEL, K-562, and U937 cell lines	N.A.	[125]
NNC 55-0396	Ovarian cancer	HO8910, A2780 cell lines	N.A.	[125]
NNC 55-0396	Ovarian cancer	Xenograft nude mice	Not reported	[125]
Paclitaxel (+/−Nickel)	Prostate cancer	LNCaP cell line	N.A.	[125]
Penfluoridol	Breast cancer, glioblastoma, pancreatic cancer, lung cancer, colon cancer	MDA-MB-231, HCC 1806, 4 TI, GBM 43, GBM 10, GBM 44, GBM 28, GBM 14, T98G, U251 MG, U87MG, SJ-GBM2, CHLA-200, Panc-1, AsPC-1, BxPC-3m LCC, LL/2, CT26 cell lines	N.A.	[133]
Pimozide	Retinoblastoma, breast cancer, glioma	Y79, WERI-Rb1 retinoblastoma, MCF7 breast cancer, C6 glioma cell lines	N.A.	[25]
TH-1177	Prostate cancer	Mice inoculated with PC3 cell line	No obvious toxicity, either in grossly or on histological examination.	[25]
Thapsigargin (+/−Nickel)	Prostate cancer	LNCaP cell line	N.A.	[125]
TTA-P2	Glioma	GIC, GliNS1, G179NS, and G166NS, U3NNN-MG cell lines	N.A.	[125]
TTA-P2	Glioma	Xenograft injection	Not reported	[125]

N.A. = Not Available due to the preclin.

## Data Availability

Not applicable.

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
