# Peer review of "T-Type Calcium Channels: A Mixed Blessing"

_ijms, 2022, doi:10.3390/ijms23179894_

Round 1

Reviewer 1 Report

Dear authors,

Your review gives interesting information about T-type calcium channels. I found that the article deals with an interesting topic and offers the readers a comprehensive overview. I have some minor comments:

1.     A table with the drugs that is known or are proposed to act on T-type calcium channels could help readers to have an overview.

2.     I found that the chapter of about cardiotoxicity is not well integrated in the manuscript. Are there any potential drugs acting on T-type calcium channels that can help improving these symptoms?

3.     Is there any preliminary clinical evidence (trials, observational studies) about repurposing drugs targeting T-type calcium channels in cancer patients?

4.     Other articles found that also sodium channels can regulate cancer cell invasion. What is known about the relationship between these two channel in cancer?

Reviewer 2 Report

Melgari and co-workers review a general property of T-type Ca2+ channel and then the channel’s role in cancer cells.  Although this review article seems to be well written, there are several points that should be addressed and may serve to amend this manuscript, as follows:

1.     It will be better to summarize in one Table an involvement of T-type Ca2+ channels in cancer, chemotherapy using T-type Ca2+ channel-related drugs and side effects of this chemotherapy.

2.     A list of abbreviations should be given to make this review easier for non-experts to read.

3.     Line 38: not “make” but “makes”?

4.     Line 65: please explain “IQ motif” shortly.

5.     Line 78: “2+” in “Ca2+” should be superscript.

6.     Line 85: “current ICaT)” should be “current (ICaT)”.

7.     Line 121: it is unlikely that T-type current is sensitive to capsaicin.  “T-rich” cells seem to respond to capsaicin.  Please check ref. [32].

8.     Line 154: not “taxane” but “taxanes”?

9.     Line 156: is “to epothilones, they” OK?  Please check English.

10.  Line 216: “hERG” should be defined in this line but not line 351.

11.  Line 309: please explain “BRAF” shortly.

12.  Line 344: “ICaL” does not seem to be defined.  Please amend this point.

13.  Line 351: not “human ether-a-go-go” but “human ether-a-go-go related gene”?

14.  Line 355: not “HIF1 and 2” but “HIF1 and 2 expressions”?

15.  Lines 358 and 361: not “NNC-55-0396” but “NNC 55-0396”?

16.  Line 364: is “resulted toxic” OK?  Please check English.

17.  Line 367: not “NNC55-0396” but “NNC 55-0396”?

18.  Line 368: please expand “UPR”.

19.  Line 371: is “selectivity approximately” OK?  Please check English.

20.  Line 378: is “a Kd 4-fold higher” OK?  Please check English.  Not “T-type” but “T-type Ca2+ channels”.  Please expand “Kd”.

21.  Line 384: “higher” or “lower” before “affinity”?

22.  Line 386: is “a process the basis” OK?  Please check English.

23.  Line 416: not “T-type” but “T-type Ca2+ channels”.

24.  References: “2+” in “Ca2+” should be superscript throughout the references.  Is the usage of “et al. of the authors” OK?  References should be presented in a unified format.  Moreover, please check all of the references whether they are cited correctly.

25.  There may be more mistakes than pointed out above.  Please check the manuscript very carefully.
